# Real-World Study: Hybrid Immunity against SARS-CoV-2 Influences the Antibody Levels and Persistency Lasting More than One Year

**DOI:** 10.3390/vaccines11111693

**Published:** 2023-11-07

**Authors:** Sitthichai Kanokudom, Jira Chansaenroj, Suvichada Assawakosri, Nungruthai Suntronwong, Ritthideach Yorsaeng, Lakkhana Wongsrisang, Ratchadawan Aeemjinda, Preeyaporn Vichaiwattana, Sirapa Klinfueng, Thaksaporn Thatsanathorn, Sittisak Honsawek, Yong Poovorawan

**Affiliations:** 1Center of Excellence in Clinical Virology, Department of Pediatrics, Faculty of Medicine, Chulalongkorn University, Bangkok 10330, Thailand; sitthichai.k@chula.ac.th (S.K.); jira.c@chula.ac.th (J.C.); 6273005030@student.chula.ac.th (S.A.); nungruthai.s@chula.ac.th (N.S.); ritthideach.yor@gmail.com (R.Y.); lakkhana4118@gmail.com (L.W.); aeemjinda.r@gmail.com (R.A.); preeya_teiy@hotmail.com (P.V.); sirapa.klinfueng@gmail.com (S.K.); thaksapohnl@hotmail.com (T.T.); 2Center of Excellence in Osteoarthritis and Musculoskeleton, Faculty of Medicine, Chulalongkorn University, King Chulalongkorn Memorial Hospital, Thai Red Cross Society, Bangkok 10330, Thailand; 3Fellow of the Royal Society of Thailand (FRS [T]), The Royal Society of Thailand, Sanam Sueapa, Dusit, Bangkok 10300, Thailand

**Keywords:** severe acute respiratory virus 2 (SARS-CoV-2), persistent immunity, hybrid immunity, long-term follow-up, durability

## Abstract

This study investigated the impact of hybrid immunity on antibody responses in the participants who received two to seven doses of the COVID-19 vaccine. The study was conducted between April and June 2023. Out of 771 serum samples analyzed, 71.7% exhibited hybrid immunity (positive for total anti-N Ig), while 28.3% showed vaccine-induced immunity (negative for total anti-N Ig). Participants were categorized based on the number of vaccine doses: 2, 3, 4, and ≥5. The findings highlight a trend where a higher number of vaccine doses received was associated with a lower infection rate. There was no significant difference in total RBD Ig levels between those who received 3, 4, or ≥5 doses in both the hybrid immunity and vaccination alone groups across all observed durations as follows: <6 months, 6 to <9 months, 9 to <12 months, and ≥12 months. Hybrid immunity consistently maintained higher total RBD Ig levels and durability compared to vaccination alone, with estimated half-lives (T_1/2_) of 189.5 days versus 106.8 days for vaccine alone. This investigation underscored the potential benefit of hybrid immunity and raised questions about the optimal strategies for further vaccine dosing.

## 1. Introduction

In December 2019, a novel beta-coronavirus named severe acute respiratory virus 2 (SARS-CoV-2) emerged in Wuhan, China. These infections led to the development of coronavirus disease 2019 (COVID-19) and quickly spread worldwide within a short period. SARS-CoV-2 infections have been linked to a broad range of clinical outcomes, ranging from asymptomatic or mild cases to severe illnesses requiring hospitalization [1]. A systematic review comprising 41 studies from 17 countries, conducted before May 2020 and involving a cumulative total of 50,155 COVD-19 confirmed patients, revealed that around 15.6% exhibited no symptoms, whereas 48.9% initially showed no symptoms but eventually developed them [2].

In April 2020, the first COVID-19 vaccine, CoronaVac (Sinovac Biotech, Beijing, China), was developed using a whole inactivated vaccine based on the Wuhan-H1 strain. Subsequently, recombinant vaccines such as viral vectors, protein subunits, and mRNA-based vaccines gained emergency use approval and were introduced in many countries. Despite using various vaccine platforms and administering multiple doses to mitigate severity, the vaccines might not entirely prevent infections [3,4]. Numerous studies have reported a high incidence of breakthrough infections due to immune-evading Omicron variants [5,6] and waning immunity over time. Waning immunity requires booster doses to re-establish high protective antibodies against infections [7,8,9]. Moreover, pre-existing immunity and breakthrough infections have also contributed to elevated antibody levels, durability, and vaccine effectiveness against infection and disease severity [3,10,11]. However, information about long-term immunity lasting more than a year influenced by hybrid immunity has not been fully elucidated. Therefore, investigating antibody levels helps determine the need for booster shots and facilitates informed vaccine management and supply decisions.

Real time-reverse transcriptase polymerase chain reaction (RT-PCR) has long served as the gold standard for confirming COVID-19 infection in laboratories. Therefore, SARS-CoV-2 antibody testing has emerged as a valuable tool. It is particularly beneficial in diagnosing and providing evidence of disease, especially in cases where RT-PCR results are negative or for asymptomatic individuals. The previous study indicated that positive detection of anti-nucleocapsid antibodies could persist for up to 270 days following a positive RT-PCR confirmation in cases of natural infection [12]. Therefore, in this study, we defined hybrid immunity in participants based on their serostatus for anti-nucleoprotein antibodies.

This study aimed to determine the proportion of individuals impacted by COVID-19 based on serostatus of anti-nucleoprotein antibodies. Additionally, it sought to evaluate the waning immunity and durability affected by hybrid immunity among a large-scale population of those who reside in Bangkok, Thailand. We conducted a real-world study of individuals receiving two to seven doses of monovalent COVID-19 vaccine with or without experiencing breakthrough infection.

## 2. Materials and Methods

### 2.1. Study Designs and Participant Enrollment

A cross-sectional real-world serosurvey was conducted between April and June 2023 at the Center for Excellence in Clinical Virology, Chulalongkorn University, Bangkok. The venous blood specimens were collected from participants who were located in the Bangkok metropolitan area. This study protocol was approved by the Institutional Review Board (IRB) of the Faculty of Medicine of Chulalongkorn University (IRB 099/65).

A total of 789 adults 18 years and older who had previously been immunized with at least two doses of any of the available COVID-19 vaccines. All individuals completed a self-record questionnaire to provide information, including their sex, age, comorbidity, vaccination records, and history of infection, confirmed by a positive test result for SARS-CoV-2 through either rapid antigen testing or quantitative reverse transcription-polymerase chain reaction (qRT-PCR). Exclusions comprised individuals who were immunocompromised, including those with autoimmune diseases or malignancies, as well as individuals with medical conditions that could potentially affect the immunological study. Participants who did not complete the required information sheet and those who were unvaccinated were excluded. All the data gathered for the final analysis in this study have undergone anonymization. Written informed consent was unnecessary and, therefore, it was waived. 

The enrolled participants in this study were categorized based on the number of vaccines they received (2, 3, 4, and ≥5 doses) and the duration since their last vaccination and/or vaccination plus infection, providing the basis for four groups: (1) <6 months, (2) 6 to <9 months, (3) 9 to <12 months, and (4) ≥12 months. After further screening of the serostatus of antibodies specific to the nucleoprotein of SARS-CoV-2, participants were categorized into those with vaccine-induced immunity or those with hybrid immunity.

### 2.2. Laboratory Assessments

Serum samples were collected for the determination of binding antibody response. This involved measuring the total immunoglobulin anti-RBD of the SARS-CoV-2 spike protein (total RBD Ig) using the Elecsys^®^ SARS-CoV-2 S electrochemiluminescence immunoassay (ECLIA; Roche Diagnostics, Basel, Switzerland). Additionally, the serum was used to examine the previous infection through screening of the total immunoglobulin anti-nucleoprotein of the SARS-CoV-2 (total anti-N Ig) using Elecsys^®^ Anti-SARS-CoV-2 N ECLIA (Roche Diagnostics GmbH, Mannheim, Germany). Both tests were conducted using the Cobas e 411 system, following the manufacturer’s instructions. To interpret the results, total RBD Ig was presented as geometric mean titers (GMT) with a 95% confidence interval (CI). However, the seropositive of total anti-N Ig (N_pos_) was determined when the results were equal to or greater than 1.0 COI, whereas results below 1.0 COI were considered negative (N_neg_).

### 2.3. Statistical Analysis

Categorical age and sex analyses were performed using the Kruskal−Wallis test, followed by Dunn’s post hoc test with Bonferroni correction. Differences between GMT of total RBD Ig among different groups, categorized by the number of vaccine doses versus the duration since the last vaccination and vaccination plus infection, anti-N Ig, and occupation for each specific time point, were carried out using the Mann–Whitney U-test. Nonlinear regression and total RBD Ig half-life (T_1/2_) were calculated using a one-phase decay model. All statistical analyses and figures were performed using GraphPad Prism version 9.0 (GraphPad, San Diego, CA, USA). A *p*-value of <0.05 was considered statistically significant. 

## 3. Results

### 3.1. Demographic Characteristics of Study Participants

A total of 789 participants were initially enrolled. Out of these, 18 participants were excluded: 12 had not completed the required information sheet for enrollment, 2 had never received any vaccines, and 3 had not attended the project during the study period. Thus, the remaining 771 participants were included in this study. The data revealed that participants who received two to seven doses of the vaccine accounted for 29 (3.8%), 134 (17.4%), 366 (47.5%), 219 (28.4%), 21 (2.7%), and 2 (0.2%) individuals, respectively. Among these participants, 320 (41.5%) had no history of infection, while 407 (52.8%) reported experiencing single infection since the COVID-19 outbreak. Additionally, 44 (5.7%) participants reported having been infected twice since the beginning of the COVID-19 outbreak. Among the 451 infected participants, 8 (2.0%) cases reported severe clinical outcomes. All eligible samples were divided into four groups based on the duration since the last vaccination and vice versa, plus infection. Subsequently, antibody testing was performed, as illustrated in Figure 1.

Baseline demographics and characteristics of participants who underwent antibody testing, including sex, age, occupation, and the duration since their last vaccination and/or vaccination plus infection, were presented in Table 1. The four groups, classified by this duration, had the following percentages of females: (1) <6 months—79.5%, (2) 6 to <9 months—80.4%, (3) 9 to <12 months—80.5%, and (4) ≥12 months—75.0%. There were no statistical differences in sex distribution between these four groups. The median age of the four groups was reported as follows: (1) <6 months—36.5, (2) 6 to <9 months—43.0, (3) 9 to <12 months—42.0, and (4) ≥12 months—41.0. In comparison, it was found that the age of the “<6 months” group was significantly lower than that of the “6 to <9 months” group (*p*-value of 0.011). However, there were no statistically significant differences in age when comparing the “<6 months” group with the “9 to <12 months” and “≥12 months” groups.

### 3.2. An Increased Number of Vaccine Doses Corresponded to a Decrease in the Infection Rate

The individuals were categorized using total anti-N Ig serostatus and compared with their infection history. A total of 771 serum samples were analyzed, of which 71.7% (553 out of 771) were classified as hybrid immunity or N_pos_, and 28.3% (218 out of 771) as vaccine-induced immunity or N_neg_. Next, we investigated the relationship between a history of infection and total anti-N Ig. The findings ultimately confirmed the results, identifying 57.7% (445 out of 771) cases as true positives, including 401 cases of single infection and 44 cases of double infection. Additionally, 27.5% (212 out of 771) cases were confirmed as true negatives. However, 14.0% (108 out of 771) cases were classified as false positives (asymptomatic infections), and 0.8% (6 out of 771) cases were found to be false negatives (Table 2).

The findings revealed that individuals who received five or more doses exhibited a 69.0% N_pos_ rate (167 out of 242). Similarly, those who received four doses showed a 70.5% N_pos_ rate (152 out of 219), while recipients of three doses had a 75.4% N_pos_ rate (101 out of 134). Notably, individuals who received two doses experienced the highest N_pos_ rate at 93.1% (27 out of 29). This study results clearly demonstrated that increased vaccine doses corresponded to a decrease in the infection rate (Table 3). 

### 3.3. The Number of Booster Doses Does Not Increase Total RBD Ig Levels and Their Longevity

This study focused on investigating the impact of hybrid immunity on the total RBD Ig response. The total RBD Ig GMTs of participants who received 2, 3, 4, or ≥5 doses were categorized based on the duration since the last vaccination and vaccination plus infection, and these results are presented in the Appendix A, Appendix A. Considering different durations, we compared the total RBD Ig levels in individuals who received 3, 4, or ≥5 doses. Our findings indicated no significant difference in total RBD Ig GMT among the boosted group (those who received 3, 4, or ≥5 doses) for both N_pos_ and N_neg_. However, there was a statistically significant increase in total RBD Ig levels in individuals with hybrid immunity, particularly those who had received ≥5 doses, compared to those who had received only 3 doses (*p*-value of 0.025), within a duration of 9 to <12 months (Figure 2A,B). This finding demonstrated that the total RBD levels in individuals who received two doses were lower than those who received booster doses, as observed over a duration longer than 9 months (Figure 2A). In parallel, we compared the total RBD Ig levels in each duration, considering different doses of vaccines. This study compared the binding antibody levels between the N_pos_ and N_neg_ groups according to the duration since the last vaccine and/or vaccination plus infection. In the N_pos_ population, individuals who received 2, 3, 4, and ≥5 doses consistently exhibited higher total RBD Ig GMT than the N_neg_ group across all observed durations (Figure 3A,B). In both groups, the binding antibody naturally waned over time.

### 3.4. Hybrid Immunity Sustained Prolonged Binding Antibody Levels

When considering overall individuals, the total RBD Ig GMT between the N_pos_ and N_neg_ groups for a short duration (1) showed similarity (GMR: 1.1, ns). However, for a long duration, the total RBD Ig of the N_pos_ group was significantly higher than the N_neg_ group (GMR: 2.4, *p*-value of <0.001 for (2); GMR: 2.6, *p*-value of <0.001 for (3); GMR, *p*-value of <0.001 for (4)). Relative to the N_pos_ groups at different durations, it was found that the increasing GMR values over a longer duration represented the influence of hybrid immunity on the antibody levels and their durability compared to the vaccine alone group (N_neg_) (Figure 4). 

To determine whether the estimated durability of total RBD Ig was influenced by hybrid immunity, we examined the relationship between total RBD Ig levels and the days since the last exposure to the vaccine and vaccination plus infection, spanning up to approximately 600 days. As well as the data presented in Figure 4, the antibody levels influenced by hybrid immunity were notably higher than those achieved through vaccination alone. The trendline of antibody levels exhibited slow and steady progress with more remarkable persistence among individuals with hybrid immunity (T_1/2_ = 189.5 days), as compared to those with the vaccine alone (T_1/2_ = 106.8 days) (as shown in Figure 5A and Figure 5B, respectively).

### 3.5. No Association between Occupation and Binding Antibody Levels

To analyze the influence of occupation on the binding antibody, each group was divided into HCW and NHCW. The GMT of total RBD Ig levels gradually declined over time among HCW and NHCW. Furthermore, the results indicated that there was no statistically significant difference between HCW and NHCW in all durations since the last vaccination and/or vaccination plus infection (1)–(4) (Appendix A). The results suggested no association between HCW and NHCW regarding the total RBD Ig levels in this population study.

## 4. Discussion

This study presented findings from serosurveys conducted on individuals who willingly underwent blood tests to assess the antibody response to SARS-CoV-2 antigens, including total RBD Ig and total anti-N Ig. Blood samples were classified based on the time interval between sample collection and the individual’s history of vaccination and infection, extending up to one year. A total of 771 eligible participants were included in the study, mainly consisting of individuals who received four doses (47.5%), followed by five (28.4%) and three doses (17.4%), respectively. Mix-and-match vaccine regimens were employed in Thailand based on vaccine accessibility and voluntary participation [13,14]. According to this study, the primary vaccination regimen consisted of two doses, mostly using inactivated vaccines such as CoronaVac (Sinovac) or BBIBP-CorV (Sinopharm), viral-vector vaccines like AZD1222 (AstraZeneca), or a combination of CoronaVac followed by AZD1222. Meanwhile, booster doses were mainly administered using AZD1222, BNT162b2 (Pfizer), and mRNA-1273 (Moderna). We observed that the population residing in the Bangkok metropolitan area, which is the capital of Thailand, had affordably accessed the vaccine compared to the country’s general population. Moreover, they were more concerned about and interested in receiving a vaccine booster. As of March 2023, the general population consisted of 69,556,294 individuals, with 39.3%, 9.4%, and 1.5% receiving 3–5 vaccines, respectively [15]. 

This study highlighted that using anti-N serostatus as a marker provided a more accurate measure when comparing records of infection history, as many cases are asymptomatic, which could lead to underestimation. Among the 771 individuals, 71.7% were classified as N_pos_, which substantially exceeded those identified by records of infection history (58.7%). The results demonstrated that the seroprevalence of COVID-19 in Bangkok observed between April and June 2023, was close to that of the previous serosurvey conducted between October 2022 and January 2023 in Chonburi province (accounting for 73.7% of 1076 out of 1459 individuals) [16]. We anticipated that the infection rate would further rise throughout the entire population of Thailand. In this study, we noted that some individuals had received a priming regimen of two inactivated vaccines (Sinovac and Sinopharm), while others had received a combination of Sinovac and AstraZeneca. Among these individuals, their last inactivated vaccine had been administered more than 541 days ago. This suggests that the inactivated antigens may not evoke an anti-N Ig level, as it appeared to be negative. This is consistent with our previous study, which showed that 95.3% (n = 645) were seronegative of anti-N IgG after last exposure to the inactivated vaccine longer than 5 months ago [17]. Hence, we can confirm that all individuals who tested positive for N protein antibodies would be indicative of a previous infection.

This finding demonstrated that the group receiving more vaccine doses had a lower infection rate than those receiving fewer doses, potentially aligning with better health and hygiene awareness. Additionally, the study suggested that those receiving booster doses had higher antibody levels and longer-lasting immunity than those with two doses. The hybrid immunity (N_pos_) provided higher total RBD Ig levels than those achieved by vaccination alone (N_neg_). The increased immunogenicity resulting from hybrid immunity observed in this study was consistent with previous findings [18]. Furthermore, the findings indicated a more sustained presence of binding antibody levels for over 6 months and long-lasting beyond 1 year in the N_pos_ group. This coincides with our previous study conducted between January 2021 and December 2022, which covered the periods of the Alpha, Beta, Delta, and Omicron variants. It demonstrated that individuals who experienced natural infection plus received 2–4 vaccine doses maintained a more stable level of total RBD antibodies for over a year [19]. The persistency of total RBD Ig in hybrid immunity was aligned with a previous study in Israel in which anti-S1 IgG levels persisted for two to six months after the infection post-three doses [20]. Moreover, this was particularly significant, as it correlated with neutralization against infection, especially the Omicron variant, and reduced disease severity [21,22]. Previous studies that corroborated this phenomenon reported that breakthrough infection improved a broad neutralization of emergent SARS-CoV-2 Omicron sub-variants [23,24]. In this study, a comparison between HCW and NHCW did not reveal any statistically significant differences across the observed durations, including periods over a year. This contrasted with a prior study conducted from October 2022 to January 2023, which showed higher anti-RBD IgG levels among HCWs compared to the general population [16]. This study suggests that occupation may not significantly affect total anti-RBD Ig levels.

After experiencing multiple outbreaks, including the predominant Omicron, global populations became infected and developed natural and/or hybrid immunity. Even in countries with a high vaccination rate, preventing the spread of COVID-19 was impossible. Based on surveillance data in China between 16 December 2022 and 19 January 2023, it was indicated that around 90% of the population had become infected [25]. Israel showed more than 50% experienced infection, as observed in February 2023 [20]. Many countries do not recommend a booster vaccination for non-clinical risk adults who have been previously infected. However, older adults and those who are immunosuppressed should be considered for an extra booster vaccine [26,27,28]. It is crucial to emphasize that relying solely on natural infection as a policy is not advisable. Natural infection poses risks of severe acute illness and long-term post-acute symptoms (Long COVID), particularly in vulnerable individuals and those with underlying health conditions. Moreover, vaccination partially protects against Long COVID, ranging from 15% to 41% [29,30,31].

Furthermore, as of 5 May 2023, COVID-19 was no longer considered a public health emergency [32]. The endemic approach to preventing COVID-19 will correspond to other respiratory diseases, emphasizing personal hygiene, mask-wearing, and community risk assessments. Consideration of COVID-19 vaccination policy can resemble influenza vaccination campaigns, aligning with the predicted predominant strains. Again, the vaccination remains essential for effectively mitigating and managing COVID-19.

This study includes specific limitations. The patient provided information about the reported infection without documented confirmation. To determine the seroprevalence, these results may not guarantee representation of the entire population. Furthermore, observing various regions of Thailand, including urban areas, can improve accuracy. Moreover, the investigation focused primarily on long-term antibody responses. Further investigation was warranted into the cellular responses associated with hybrid and vaccine-induced immunity. We did not investigate the effects of specific vaccine types in this study. Analysis based on vaccine types has been challenged due to the diversity of vaccine regimens in the Thai population, in contrast to the United States, where mRNA vaccines are predominantly utilized. 

## 5. Conclusions

The findings emphasized the significance of considering vaccination and infection history in designing effective COVID-19 control strategies. The evidence showed that reduced infection rates were associated with increased vaccine doses. This highlights the potential benefits of strategic dosing regimens. Moreover, the prolonged and elevated antibody levels observed among individuals with hybrid immunity underscore its potential importance in providing sustained protection. The findings might suggest that individuals who have been infected can choose to receive additional booster shots annually or when it is deemed appropriate. Meanwhile, individuals who have never been infected can be re-immunized more rapidly.

## Figures and Tables

**Figure 1 vaccines-11-01693-f001:**
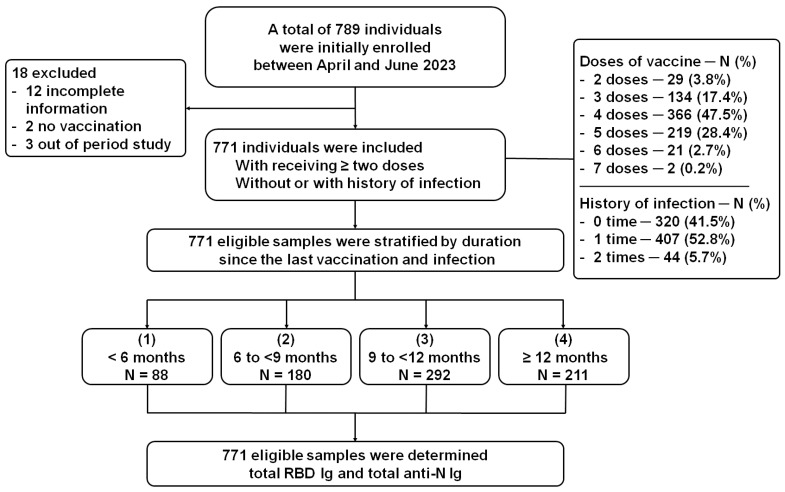
Participants flow in this study. The cross-sectional serosurvey included the participants who sought antibody testing between April and May 2023. The participants were eligible and were categorized into four groups: (1) <6 months, (2) 6 to <9 months, (3) 9 to <12 months, and (4) ≥12 months. Subsequently, all eligible samples were subsequently evaluated for the total RBD Ig and total anti-N Ig.

**Figure 2 vaccines-11-01693-f002:**
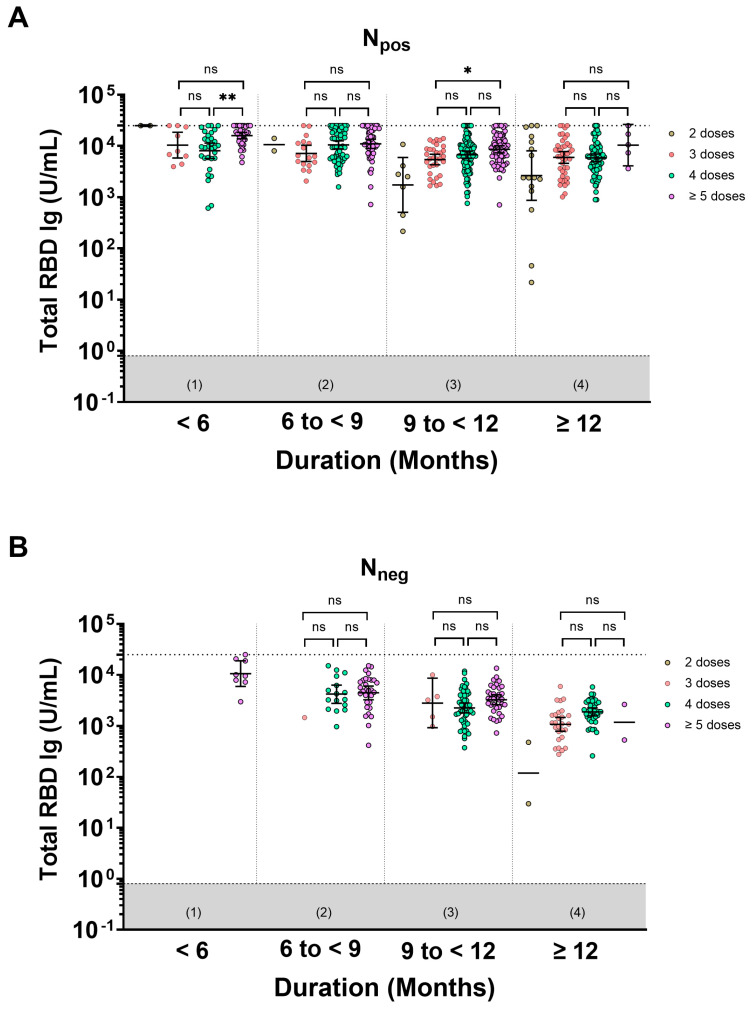
The pairwise comparisons of total RBD Ig levels among individuals who have received 3, 4, or ≥5 doses, categorized by the duration since the last vaccination and vaccination plus infection (months) as follows: (1) <6 months, (2) 6 to <9 months, (3) 9 to <12 months, and (4) ≥12 months. The individuals were divided based on serostatus of total anti-N Ig consisting of (**A**) N_pos_ and (**B**) N_neg_. Lines represent the geometric mean titer (GMT) with 95% confidence intervals (95% CI). The upper limit of the total RBD Ig is reported as 25,000 U/mL. The gray area indicates the seronegativity of total RBD Ig (<0.8 U/mL). Bars represented statistical significance set at *p* < 0.05 (*), *p* < 0.01 (**), and no statistical significance (ns).

**Figure 3 vaccines-11-01693-f003:**
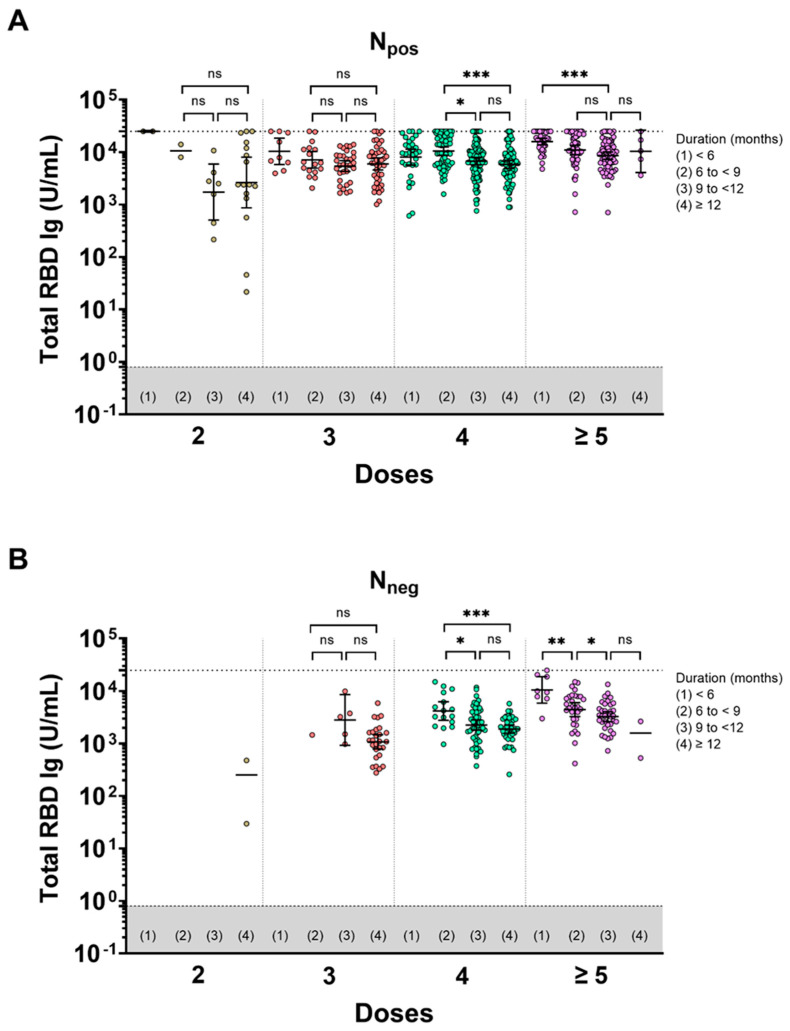
The pairwise comparisons of total RBD Ig levels in each duration since the last vaccination and vaccination plus infection (months) within the number of doses. The individuals were divided based on serostatus of total anti-N Ig consisting of (**A**) N_pos_ and (**B**) N_neg_. Lines represent the geometric mean titer (GMT) with 95% confidence intervals (95% CI). The upper limit of the total RBD Ig is reported as 25,000 U/mL. The gray area indicates the seronegativity of total RBD Ig (<0.8 U/mL). Bars represent statistical significance set at *p* < 0.05 (*), *p* < 0.01 (**), *p* < 0.001 (***), and no statistical significance (ns). Duration since last vaccination and vaccination plus infection (months) divided into four groups: (1) <6 months, (2) 6 to <9 months, (3) 9 to <12 months, and (4) ≥12 months.

**Figure 4 vaccines-11-01693-f004:**
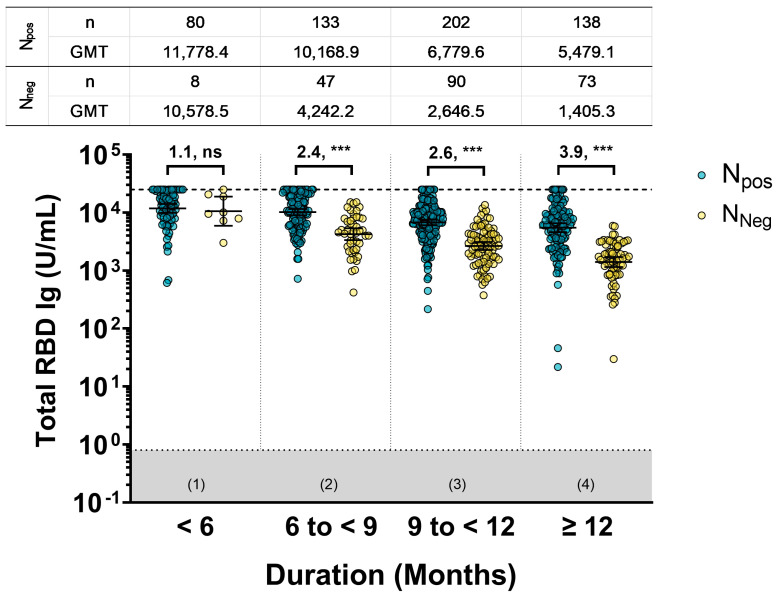
The total RBD Ig level of the participants was classified by total anti-N Ig testing (N_pos_ and N_neg_). The relationship between total RBD Ig levels (U/mL) and the duration since the last vaccination and vaccination plus infection, categorized as follows: (1) <6 months, (2) 6 to <9 months, (3) 9 to <12 months, and (4) ≥12 months. The gray area indicates the seronegativity of total RBD Ig (<0.8 U/mL). Lines represent the geometric mean titer (GMT) with 95% confidence intervals (95% CI). The upper limit of the total RBD Ig is reported as 25,000 U/mL. The gray area indicates the seronegativity of total RBD Ig (<0.8 U/mL). A pairwise comparison shows the geometric mean ratio (GMR) and statistical significance set at *p* < 0.001 (***), and no statistical significance (ns).

**Figure 5 vaccines-11-01693-f005:**
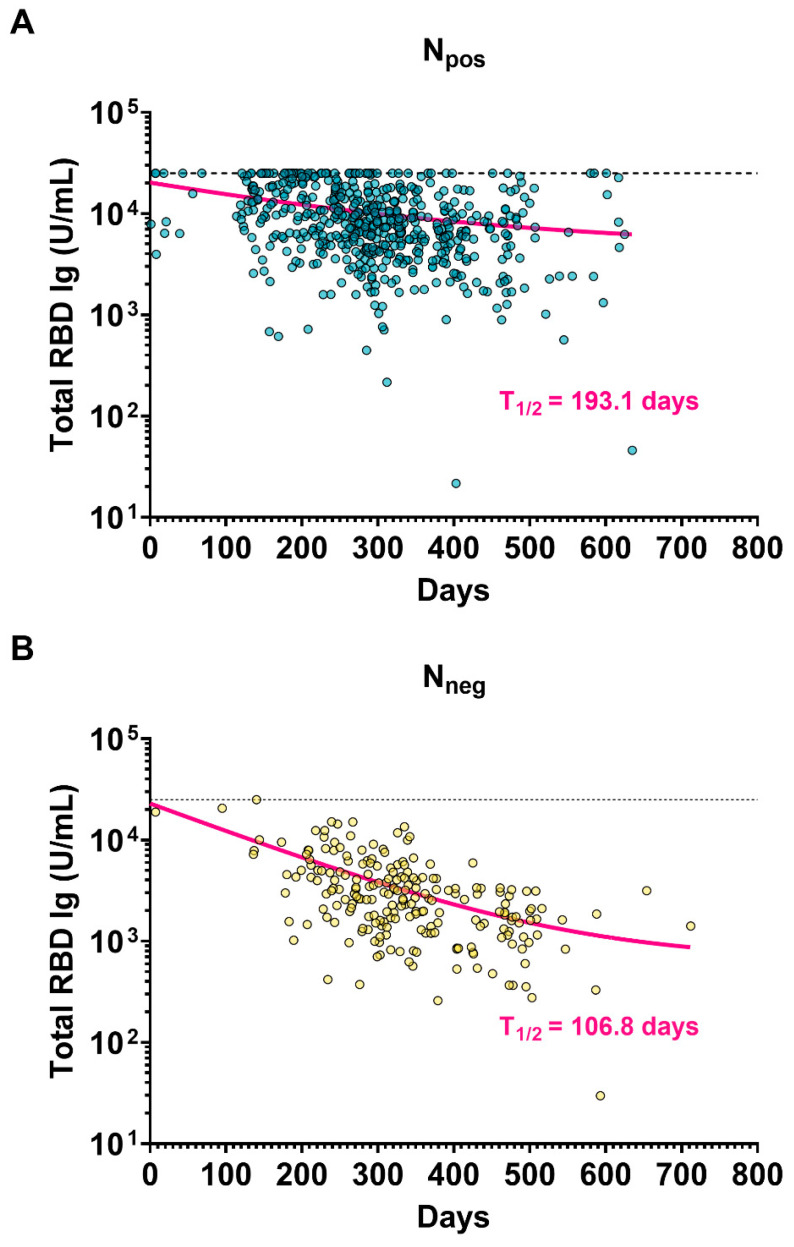
The stability of the total RBD Ig of the total participants was classified by total anti-N Ig testing (N_pos_ and N_neg_). The scatter plot of total RBD Ig is shown for (**A**) hybrid immunity or (**B**) vaccine alone, comparing the days since the last vaccination. The trendline and half-life of total RBD Ig (T_1/2_) were calculated based on a one-phase decay model.

**Table 1 vaccines-11-01693-t001:** Demographics and characteristics of the participants who have undergone antibody testing.

Characteristics	All Participants	(1) <6	(2) 6 to <9	(3) 9 to <12	(4) ≥12
Total n (%)	771	88 (11.4)	180 (23.3)	292 (37.9)	211 (27.4)
Female n (%)	608 (78.9)	70 (79.5)	144 (80.4)	235 (80.5)	159 (75.0)
Age years,					
Med	42.0	36.5	43.0	42.0	41.0
[IQR]	[30.0–52.0]	[27.0–48.8]	[33.0–56.0]	[30.0–51.0]	[30.0–52.0]
Occupation n (%)					
- HCW	267 (34.6)	45 (51.1)	53 (29.6)	94 (32.2)	75 (35.4)
- NHCW	504 (65.4)	43 (48.9)	126 (70.4)	198 (67.8)	137 (64.6)
Duration since last vaccination and/or vaccination plus infection, days					
Med	299.0	141.0	231.5	308.5	440.0
[IQR]	[242.0–378.0]	[126.3–159.5]	[207.0–250.0]	[290.0–331.0]	[397.0–481.5]

Abbreviations: Med, Median; IQR, Interquartile range; HCW, Healthcare worker; NHCW, non-healthcare worker.

**Table 2 vaccines-11-01693-t002:** The relationship between a history of infection and the presence of total anti-N Ig.

**Serostatus Classified by Total anti-N Ig**		**Infection History #**		
		**Yes**	**No**	**Total**
	**N_pos_ ^¶^**	445 (57.7%)	108 (14.0%) *	553 (71.7%)
	**N_neg_ ^¶^**	6 (0.8%) **	212 (27.5%)	218 (28.3%)
	**Total**	451 (58.5%)	320 (41.5%)	771 (100.0%)

^¶^ The serostatus of total anti-N Ig was determined when the results were equal to or greater than 1.0 COI (N_pos_), whereas results below 1.0 COI are considered negative (N_neg_). # The history of infection has been confirmed by RT-PCT and/or ATK. * The last exposure to inactivated vaccines (Sinovac and Sinopharm) is equal to or more than 584 days. Therefore, these individuals are supposed to have an asymptomatic infection. ** This participant has a confirmed history of infection (single infection); however, the serostatus is still seronegative (false positive).

**Table 3 vaccines-11-01693-t003:** The ratio of seropositive cases for total anti-N Ig to the number of vaccine doses.

The Number of Vaccine Doses	The N_pos_ Rate
≥5 doses	69.0% (167 out of 242)
4 doses	70.5% (152 out of 219)
3 doses	75.4% (101 out of 134)
2 doses	93.1% (27 out of 29)

## Data Availability

The datasets generated and analyzed during the current study are available from the corresponding author upon reasonable request.

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
