# Peer review of "Real-World Study: Hybrid Immunity against SARS-CoV-2 Influences the Antibody Levels and Persistency Lasting More than One Year"

_vaccines, 2023, doi:10.3390/vaccines11111693_

Round 1
Reviewer 1 Report
Comments and Suggestions for Authors
- Provide more context about the study participants, including demographics and the rationale for selecting healthy adults aged 18 and older who had received at least two doses of the COVID-19 vaccine. Explain how the selection criteria relate to the research objectives.
- Clearly describe the laboratory assessments, including the methods and equipment used. You may want to provide a brief rationale for choosing these specific assessments and mention any quality control measures.
- While you've mentioned the statistical tools used, it's important to outline the specific statistical tests applied for various analyses. Describe why each test was chosen and its relevance to the study's objectives
- The manuscript refers to software and equipment (IBM SPSS and GraphPad Prism), but it's advisable to provide proper citations for these tools, including version numbers, to enhance reproducibility.
- Please incorporate additional data concerning immune responses and immune evasion to accessory proteins of SARS-CoV-2 into the 'INTRODUCTION' section. You can use following articles :
1. doi.org/10.1038/s41577-022-00715-2
2. doi: 10.1016/j.biopha.2022.113889
Comments on the Quality of English LanguageMinor editing of English language required
Author Response
Response to Reviewer 1
Comments and Suggestions for Authors
- Provide more context about the study participants, including demographics and the rationale for selecting healthy adults aged 18 and older who had received at least two doses of the COVID-19 vaccine. Explain how the selection criteria relate to the research objectives.
Response: We have revised section “2.1 Study Designs and Participant Enrollment”. (line 84, 89-91, 99-101)
- Clearly describe the laboratory assessments, including the methods and equipment used. You may want to provide a brief rationale for choosing these specific assessments and mention any quality control measures.
Response: We have added more details and revised accordingly (line105-110). The test is intended as an aid to assess the adaptive humoral immune response after natural infection or vaccination. These tests have been authorized FDA EUA. Boths test have provide the calibrator control within kits.
- While you've mentioned the statistical tools used, it's important to outline the specific statistical tests applied for various analyses. Describe why each test was chosen and its relevance to the study's objectives
Response: To reduce the confusing by using two software. We have re-analyzed using only Graphpad Prism. This section has been revised.
- The manuscript refers to software and equipment (IBM SPSS and GraphPad Prism), but it's advisable to provide proper citations for these tools, including version numbers, to enhance reproducibility.
Response: The statistical analysis section has been revised.
- Please incorporate additional data concerning immune responses and immune evasion to accessory proteins of SARS-CoV-2 into the 'INTRODUCTION' section. You can use following articles :
- doi.org/10.1038/s41577-022-00715-2
- doi: 10.1016/j.biopha.2022.113889
Response: We added a few sentences into troduction (line 45-47).

Reviewer 2 Report
Comments and Suggestions for Authors
manuscript entitle "Real-world study: hybrid immunity against SARS-CoV-2 influences the antibody levels and persistency lasting more than one year" by Kanokudon et al., describe the use o Ig anti N SARS CoV-2 as biomarker to differentiate subjects who were vaccinated with and without breakthrough infection; as described previously by Navaratnam et al. (2022). Kanokudan et al concluded that hybrid immunity provided higher Ig to RBD; and the more vaccination, the longer IgG RBD durability and titre maintained.
The value of the findings may be increased within sections
Metholodology: how did authors determined past and how many times infection occurs of SARS CoV-2 among subjects that showed in table 2 should be described more detailed.
Results: it is rather difficult to read the differences between figure 2 and 3; further, to confirm the data please add supplementary tables describing the numbers ; while describing clearer in the result section.
Discussion suggested that inactivation antigen did not evoke Ig response to N antigen; what would it be the reason and supporting reference. Please add to discussion.
Author Response
Response to Reviewer 2
Comments and Suggestions for Authors
manuscript entitle "Real-world study: hybrid immunity against SARS-CoV-2 influences the antibody levels and persistency lasting more than one year" by Kanokudon et al., describe the use of Ig anti N SARS CoV-2 as biomarker to differentiate subjects who were vaccinated with and without breakthrough infection; as described previously by Navaratnam et al. (2022). Kanokudan et al concluded that hybrid immunity provided higher Ig to RBD; and the more vaccination, the longer IgG RBD durability and titre maintained.
The value of the findings may be increased within sections
Metholodology: how did authors determined past and how many times infection occurs of SARS CoV-2 among subjects that showed in table 2 should be described more detailed.
Response: All participants filled out a questionnaire providing information about their infection history on the day of blood sampling. Consequently, we analyzed the data based on their infection history and the serotesting results, as shown in Table 2. We also add more context describing the history of infection comprising single and double infected cases (line 165-166, 175).
Results: it is rather difficult to read the differences between figure 2 and 3; further, to confirm the data please add supplementary tables describing the numbers ; while describing clearer in the result section.
Response: We have included a new supplementary Table S1 that provides information on the number of participants and the geometric mean titers (GMT) for the groups represented in Figures 2 and 3. We also and more context in line 187-189
Discussion suggested that inactivation antigen did not evoke Ig response to N antigen; what would it be the reason and supporting reference. Please add to discussion.
Response: We have addressed the reason why we believe that the anti-N result was not interfere by inactivated vaccine. (line 294-302)

Reviewer 3 Report
Comments and Suggestions for Authors
Review Report for article: Real-world study: hybrid immunity against SARS-CoV-2 influences the antibody levels and persistency lasting more than one 3 year
A brief summary
The aim of this article was to determine the proportion of individuals impacted by COVID-19 based on serostatus of anti-nucleoprotein antibodies. Additionally, it sought to evaluate the waning immunity and durability affected by hybrid immunity among a large-scale population of those who reside in Bangkok, Thailand. A real-world study of individuals receiving two to seven doses of monovalent COVID-19 vaccine with or without experiencing the breakthrough infection was conducted.
General concept comments
Article: highlighting areas of weakness, the testability of the hypothesis, methodological inaccuracies, missing controls, etc.
The article presents an extensive data on an innovative novel area to do with SARS-COV-2.
Review: commenting on the completeness of the review topic covered, the relevance of the review topic, the gap in knowledge identified, the appropriateness of references, etc.
These comments are focused on the scientific content of the manuscript and should be specific enough for the authors to be able to respond.
The article presents exhaustive data on the subject matter.
Specific comments referring to line numbers, tables or figures that point out inaccuracies within the text or sentences that are unclear. These comments should also focus on the scientific content and not on spelling, formatting or English language problems, as these can be addressed at a later stage by our internal staff.
The article is a significant contribution to Journal Vaccine. It should be accepted for publication. Given its strength of scientific data.
Kindly observe the Journal format

Easy to read and comprehend.
Author Response
Response to Reviewer 3
Comments and Suggestions for Authors
Review Report for article: Real-world study: hybrid immunity against SARS-CoV-2 influences the antibody levels and persistency lasting more than one year
A brief summary
The aim of this article was to determine the proportion of individuals impacted by COVID-19 based on serostatus of anti-nucleoprotein antibodies. Additionally, it sought to evaluate the waning immunity and durability affected by hybrid immunity among a large-scale population of those who reside in Bangkok, Thailand. A real-world study of individuals receiving two to seven doses of monovalent COVID-19 vaccine with or without experiencing the breakthrough infection was conducted.
General concept comments
Article: highlighting areas of weakness, the testability of the hypothesis, methodological inaccuracies, missing controls etc.
The article presents an extensive data on an innovative novel area to do with SARS-COV-2.
Review: commenting on the completeness of the review topic covered, the relevance of the review topic, the gap in knowledge identified, the appropriateness of references, etc.
These comments are focused on the scientific content of the manuscript and should be specific enough for the authors to be able to respond.
The article presents exhaustive data on the subject matter.
Specific comments referring to line numbers, tables or figures that point out inaccuracies within the text or sentences that are unclear. These comments should also focus on the scientific content and not on spelling, formatting or English language problems, as these can be addressed at a later stage by our internal staff.
The article is a significant contribution to Journal Vaccine. It should be accepted for publication. Given its strength of scientific data.
Response: We are grateful for the valuable feedback.
Kindly observe the Journal format
